# PlanT: Explainable Planning Transformers via Object-Level Representations

**Katrin Renz**[1,2]    **Kashyap Chitta**[1,2]    **Otniel-Bogdan Mercea**[1]
**A. Sophia Koepke**[1]    **Zeynep Akata**[1,2,3]    **Andreas Geiger**[1,2]
[1]University of Tübingen        [2]Max Planck Institute for Intelligent Systems, Tübingen
[3]Max Planck Institute for Informatics, Saarbrücken
https://www.katrinrenz.de/plant

**Abstract:** Planning an optimal route in a complex environment requires efficient reasoning about the surrounding scene. While human drivers prioritize important objects and ignore details not relevant to the decision, learning-based planners typically extract features from dense, high-dimensional grid representations containing all vehicle and road context information. In this paper, we propose PlanT, a novel approach for planning in the context of self-driving that uses a standard transformer architecture. PlanT is based on imitation learning with a compact object-level input representation. On the Longest6 benchmark for CARLA, PlanT outperforms all prior methods (matching the driving score of the expert) while being $5.3\times$ faster than equivalent pixel-based planning baselines during inference. Combining PlanT with an off-the-shelf perception module provides a sensor-based driving system that is more than 10 points better in terms of driving score than the existing state of the art. Furthermore, we propose an evaluation protocol to quantify the ability of planners to identify relevant objects, providing insights regarding their decision-making. Our results indicate that PlanT can focus on the most relevant object in the scene, even when this object is geometrically distant.

**Keywords:** Autonomous Driving, Transformers, Explainability

## 1 Introduction

The ability to plan is an important aspect of human intelligence, allowing us to solve complex navigation tasks. For example, to change lanes on a busy highway, a driver must wait for sufficient space in the new lane and adjust the speed based on the expected behavior of the other vehicles. Humans quickly learn this and can generalize to new scenarios, a trait we would also like autonomous agents to have. Due to the difficulty of the planning task, the field of autonomous driving is shifting away from traditional rule-based algorithms [1, 2, 3, 4, 5, 6, 7, 8] towards learning-based solutions [9, 10, 11, 12, 13, 14]. Learning-based planners directly map the environmental state representation (e.g., HD maps and object bounding boxes) to waypoints or vehicle controls. They emerged as a scalable alternative to rule-based planners which require significant manual effort to design.

Interestingly, while humans reason about the world in terms of objects [15, 16, 17], most existing learned planners [9, 12, 18] choose a high-dimensional pixel-level input representation by rendering bird's eye view (BEV) images of detailed HD maps (Fig. 1 left). It is widely believed that this kind of accurate scene understanding is key for robust self-driving vehicles, leading to significant interest in recovering pixel-level BEV information from sensor inputs [19, 20, 21, 22, 23, 24]. In this paper, we investigate whether such detailed representations are actually necessary to achieve convincing planning performance. We propose PlanT, a learning-based planner that leverages an object-level representation (Fig. 1 right) as an input to a transformer encoder [25]. We represent a scene as a set of features corresponding to (1) nearby vehicles and (2) the route the planner must follow. We show that despite the low feature dimensionality, our model achieves state-of-the-art results. We then propose a novel evaluation scheme and metric to analyze explainability which is generally applicable to any learning-based planner. Specifically, we test the ability of a planner to identify the objects that are the most relevant to account for to plan a collision-free route.

6th Conference on Robot Learning (CoRL 2022), Auckland, New Zealand.

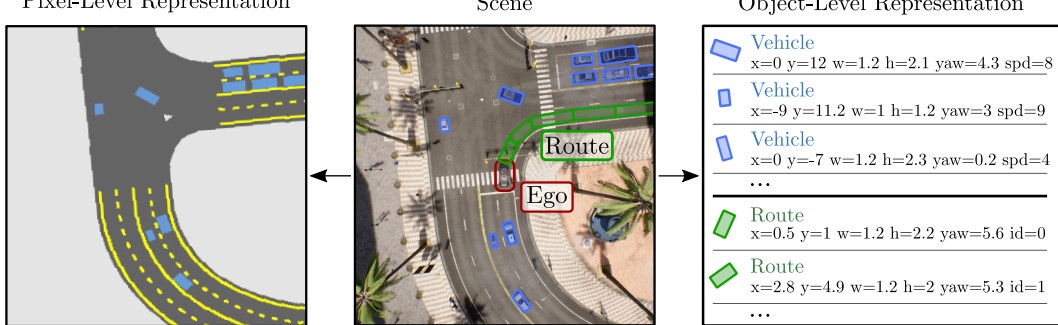

Figure 1: **Scene Representations for Planning.** As an alternative to the dominant paradigm of pixel-level planners (left), we show the effectiveness of compact object-level representations (right).

We perform a detailed empirical analysis of learning-based planning on the *Longest6* benchmark [26] of the CARLA simulator [27]. We first identify the key missing elements in the design of existing learned planners such as their incomplete field of view and sub-optimal dataset and model sizes. We then show the advantages of our proposed transformer architecture, including improvements in performance and significantly faster inference times. Finally, we show that the attention weights of the transformer, which are readily accessible, can be used to represent object relevance. Our qualitative and quantitative results on explainability confirm that PlanT attends to the objects that match our intuition for the relevance of objects for safe driving.

**Contributions.** (1) Using a simple object-level representation, we significantly improve upon the previous state of the art for planning on CARLA via PlanT, our novel transformer-based approach. (2) Through a comprehensive experimental study, we identify that the ego vehicle's route, a full 360° field of view, and information about vehicle speeds are critical elements of a planner's input representation. (3) We propose a protocol and metric for evaluating a planner's prioritization of obstacles in a scene and show that PlanT is more explainable than CNN-based methods, i.e., the attention weights of the transformer identify the most relevant objects more reliably.

## 2   Related Work

**Intermediate Representations for Driving.** Early work on decoupling end-to-end driving into two stages predicts a set of low-dimensional *affordances* from sensor inputs with CNNs which are then input to a rule-based planner [28]. These affordances are scene-descriptive attributes (e.g. emergency brake, red light, center-line distance, angle) that are compact, yet comprehensive enough to enable simple driving tasks, such as urban driving on the initial version of CARLA [27]. Unfortunately, methods based on affordances perform poorly on subsequent benchmarks in CARLA which involve higher task complexity [29]. Most state-of-the-art driving models instead rely heavily on annotated 2D data either as intermediate representations or auxiliary training objectives [26, 30]. Several subsequent studies show that using semantic segmentation as an intermediate representation helps for navigational tasks [31, 32, 33, 34]. More recently, there has been a rapid growth in interest in using BEV semantic segmentation maps as the input representation to planners [9, 12, 30, 18]. To reduce the immense labeling cost of such segmentation methods, Behl et al. [35] propose visual abstractions, which are label-efficient alternatives to dense 2D semantic segmentation maps. They show that reduced class counts and the use of bounding boxes instead of pixel-accurate masks for certain classes are sufficient. Wang et al. [36] explore the use of object-centric representations for planning by explicitly extracting objects and rendering them into a BEV input for a planner. However, so far, the literature lacks a systematic analysis of whether object-centric representations are better or worse than BEV context techniques for planning in dense traffic, which we address in this work. We keep our representation simple and compact by directly considering the set of objects as inputs to our models. In addition to baselines using CNNs to process the object-centric representation, we show that using a transformer leads to improved performance, efficiency, and explainability.

**Transformers for Forecasting.** Transformers obtain impressive results in several research areas [25, 37, 38, 39], including simple interactive environments such as Atari games [40, 41, 42, 43, 44]. While the end objective differs, one application domain that involves similar challenges

to planning is motion forecasting. Most existing motion forecasting methods use a rasterized input in combination with a CNN-based network architecture [45, 46, 47, 48, 49, 50]. Gao et al. [51] show the advantages of object-level representations for motion forecasting via Graph Neural Networks (GNN). Several follow-ups to this work use object-level representations in combination with Transformer-based architectures [52, 53, 54]. Our key distinctions when compared to these methods are the architectural simplicity of PlanT (our use of simple self-attention transformer blocks and the proposed route representation) as well as our closed-loop evaluation protocol (we evaluate the driving performance in simulation and report online driving metrics).

**Explainability.** Explaining the decisions of neural networks is a rapidly evolving research field [55, 56, 57, 58, 59, 60, 61]. In the context of self-driving cars, existing work uses text [62] or heatmaps [63] to explain decisions. In our work, we can directly obtain post hoc explanations for decisions of our learning-based PlanT architecture by considering its learned attention. While the concurrent work CAPO [64] uses a similar strategy, it only considers pedestrian-ego interactions on an empty route, while we consider the full planning task in an urban environment with dense traffic. Furthermore, we introduce a simple metric to measure the quality of explanations for a planner.

## 3 Planning Transformers

In this section, we provide details about our task setup, novel scene representation, simple but effective architecture, and training strategy resulting in state-of-the-art performance. A PyTorch-style pseudo-code snippet outlining PlanT and its training is provided in the supplementary material.

**Task.** We consider the task of point-to-point navigation in an urban setting where the goal is to drive from a start to a goal location while reacting to other dynamic agents and following traffic rules. We use Imitation Learning (IL) to train the driving agent. The goal of IL is to learn a policy $\pi$ that imitates the behavior of an expert $\pi^*$ (the expert implementation is described in Section 4). In our setup, the policy is a mapping $\pi : \mathcal{X} \to \mathcal{W}$ from our novel object-level input representation $\mathcal{X}$ to the future trajectory $\mathcal{W}$ of an expert driver. For following traffic rules, we assume access to the state of the next traffic light relevant to the ego vehicle $l \in \{\text{green}, \text{red}\}$.

**Tokenization.** To encode the task-specific information required from the scene, we represent it using a set of objects, with vehicles and segments of the route each being assigned an oriented bounding box in BEV space (Fig. 1 right). Let $\mathcal{X}_t = \mathcal{V}_t \cup \mathcal{S}_t$, where $\mathcal{V}_t \in \mathbb{R}^{V_t \times A}$ and $\mathcal{S}_t \in \mathbb{R}^{S_t \times A}$ represent the set of vehicles and the set of route segments at time-step $t$ with $A = 6$ attributes each. Specifically, if $\mathbf{o}_{i,t} \in \mathcal{X}_t$ represents a particular object, the attributes of $\mathbf{o}_{i,t}$ include an object type-specific attribute $z_{i,t}$ (described below), the position of the bounding box $(x_{i,t}, y_{i,t})$ relative to the ego vehicle, the orientation $\varphi_{i,t} \in [0, 2\pi]$, and the extent $(w_{i,t}, h_{i,t})$. Thus, each object $\mathbf{o}_{i,t}$ can be described as a vector $\mathbf{o}_{i,t} = \{z_{i,t}, x_{i,t}, y_{i,t}, \varphi_{i,t}, w_{i,t}, h_{i,t}\}$, or concisely as $\{\mathbf{o}_{i,t,a}\}_{a=1}^{6}$.

For the vehicles $\mathcal{V}_t$, we extract the attributes directly from the simulator in our main experiments and use an off-the-shelf perception module based on CenterNet [65] (described in the supplementary material) for experiments involving a full driving system. We consider only vehicles up to a distance $D_{max}$ from the ego vehicle, and use $\mathbf{o}_{i,t,1}$ (i.e., $z_{i,t}$) to represent the speed.

To obtain the route segments $\mathcal{S}_t$, we first sample a dense set of $N_t$ points $\mathcal{U}_t \in \mathbb{R}^{N_t \times 2}$ along the route ahead of the ego vehicle at time-step $t$. We directly use the ground-truth points from CARLA as $\mathcal{U}_t$ in our main experiments and predict them with a perception module for the PlanT with perception experiments in Section 4.2. The points are subsampled using the Ramer-Douglas-Peucker algorithm [66, 67] to select a subset $\hat{\mathcal{U}}_t$. One segment spans the area between two points subsampled from the route, $\mathbf{u}_{i,t}, \mathbf{u}_{i+1,t} \in \hat{\mathcal{U}}_t$. Specifically, $\mathbf{o}_{i,t,1}$ (i.e., $z_{i,t}$) denotes the ordering for the current time-step $t$, starting from 0 for the segment closest to the ego vehicle. We set the segment length $\mathbf{o}_{i,t,6} = ||\mathbf{u}_{i,t} - \mathbf{u}_{i+1,t}||_2$, and the width, $\mathbf{o}_{i,t,5}$, equal to the lane width. In addition, we clip $\mathbf{o}_{i,t,6} \leq L_{max}, \forall i, t$; and always input a fixed number of segments $N_s$ to our policy. More details and visualizations of the route representation are provided in the supplementary material.

**Token Embeddings.** Our model is illustrated in Fig. 2. As a first step, applying a transformer backbone requires the generation of embeddings for each input token, for which we define a linear projection $\rho : \mathbb{R}^6 \to \mathbb{R}^H$ (where $H$ is the desired hidden dimensionality). To obtain token embeddings $\mathbf{e}_{i,t}$, we add the projected input tokens $\mathbf{o}_{i,t}$ to a learnable object type embedding vector $\mathbf{e}_v \in \mathbb{R}^H$ or $\mathbf{e}_s \in \mathbb{R}^H$, indicating to which type the token belongs (vehicle or route segment).

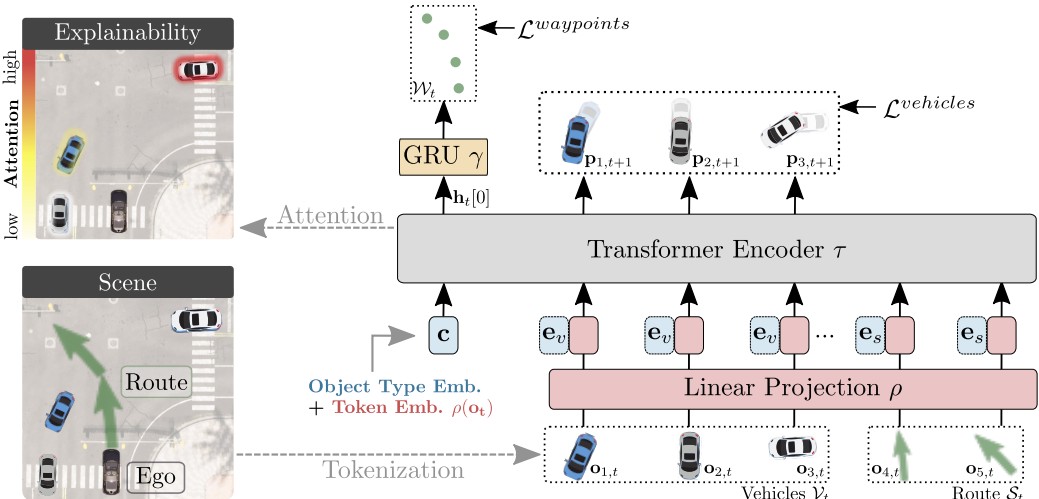

Figure 2: **Planning Transformer (PlanT).** We represent a scene (bottom left) using a set of objects containing the vehicles and route to follow (green arrows). We embed these via a linear projection (bottom right) and process them with a transformer encoder. PlanT outputs future waypoints with a GRU decoder. We use a self-supervised auxiliary task of predicting the future of other vehicles. Further, extracting and visualizing the attention weights yields an explainable decision (top left).

$$\mathbf{e}_{i,t} = \begin{cases} \rho(\mathbf{o}_{i,t}) + \mathbf{e}_v & \forall \quad \mathbf{o}_{i,t} \in \mathcal{V}_t, \\ \rho(\mathbf{o}_{i,t}) + \mathbf{e}_s & \forall \quad \mathbf{o}_{i,t} \in \mathcal{S}_t. \end{cases} \tag{1}$$

**Main Task: Waypoint Prediction.** The main building block for the IL policy $\pi$ is a standard transformer encoder, $\tau : \mathbb{R}^{V_t+S_t+1 \times H} \to \mathbb{R}^{V_t+S_t+1 \times H}$, based on the BERT architecture [37]. Specifically, we define a learnable [CLS] token, $\mathbf{c} \in \mathbb{R}^H$ (based on [37, 39]) and stack this with other token embeddings to obtain the transformer input $[\mathbf{c}, \mathbf{e}_{1,t}, ..., \mathbf{e}_{V_t+S_t,t}]$. The [CLS] token's processing through $\tau$ involves an attention-based aggregation of the features from all other tokens, after which it is used for generating the waypoint predictions via an auto-regressive waypoint decoder, $\gamma : \mathbb{R}^{(H+1)} \to \mathbb{R}^{W \times 2}$. For a detailed description of the waypoint decoder architecture, see [18, 26]. We concatenate the binary traffic light flag, $l_t$ to the transformer output as the initial hidden state to the decoder which makes use of GRUs [68] to predict the future trajectory $\mathcal{W}_t$ of the ego vehicle, centered at the coordinate frame of the current time-step $t$. The trajectory is represented by a sequence of 2D waypoints in BEV space, $\{\mathbf{w}_w = (x_w, y_w)\}_{w=t+1}^{t+W}$ for $W = 4$ future time-steps:

$$\mathcal{W}_t = \gamma(l_t, \mathbf{h}_t[0]), \quad \text{where } \mathbf{h}_t = \tau([\mathbf{c}, \mathbf{e}_{1,t}, ..., \mathbf{e}_{V_t+S_t,t}]). \tag{2}$$

**Auxiliary Task: Vehicle Future Prediction.** In addition to the primary waypoint prediction task, we propose the auxiliary task of predicting the future attributes of other vehicles. This is aligned with the overall driving goal in two ways. (1) The ability to reason about the future of other vehicles is important in an urban environment as it heavily influences the ego vehicle's own future. (2) Our main task is to predict the ego vehicle's future trajectory, which means the output feature of the transformer needs to encode all the information necessary to predict the future. Supervising the outputs of all vehicles on a similar task (i.e., predicting vehicle poses at a future time-step) exploits synergies between the task of the ego vehicle and the other vehicles [69, 30]. Specifically, using the output embeddings $\{\mathbf{h}_{i,t}\}_{i=1}^{V_t}$ corresponding to all vehicle tokens $\mathbf{o}_{i,t} \in \mathcal{V}_t$, we predict class probabilities $\{\{\mathbf{p}_{i,t+1,a}\}_{a=1}^6\}_{i=1}^{V_t}$ for the speed, position, orientation, and extent attributes from the next time-step $\{\mathbf{o}_{i,t+1,a}\}_{a=1}^6$ using a linear layer per attribute type $\{\psi_a : \mathbb{R}^H \to \mathbb{R}^{Z_a}\}_{a=1}^6$:

$$\mathbf{p}_{i,t+1,a} = \text{Softmax}(\psi_a(\mathbf{h}_{i,t})), \quad \text{where } a = 1, ..., 6. \tag{3}$$

We choose to discretize each attribute into $Z_a$ bins to allow for uncertainty in the predictions since the future is multi-modal. This is also better aligned with how humans drive without predicting exact locations and velocities, where a rough estimate is sufficient to make a safe decision.

**Loss Functions.** Following recent driving models [9, 11, 30, 26], we leverage the $L_1$ loss to the ground truth future waypoints $\mathbf{w}^{gt}$ as our main training objective. For the auxiliary task, we calculate

the cross-entropy loss $\mathcal{L}_{CE}$ using a one-hot encoded representation $\mathbf{p}_{i,t+1}^{gt}$ of the ground truth future vehicle attributes $\mathbf{o}_{i,t+1}^{gt}$. We train the model in a multi-task setting using a weighted combination of these losses with a weighting factor $\lambda$:

$$\mathcal{L} = \underbrace{\frac{1}{W} \sum_{w=1}^{W} ||\mathbf{w}_w - \mathbf{w}_w^{gt}||_1}_{\mathcal{L}^{waypoints}} + \underbrace{\frac{\lambda}{V} \sum_{i=1}^{V_t} \sum_{a=1}^{6} \mathcal{L}_{CE}\left(\mathbf{p}_{i,t+1,a}, \mathbf{p}_{i,t+1,a}^{gt}\right)}_{\mathcal{L}^{vehicles}}. \tag{4}$$

## 4  Experiments

In this section, we describe our experimental setup, evaluate the driving performance of our approach, analyze the explainability of its driving decisions, and finally discuss limitations.

**Dataset and Benchmark.** We use the expert, dataset and evaluation benchmark *Longest 6* proposed by [26]. The expert policy is a rule-based algorithm with access to ground truth locations of the vehicles as well as privileged information that is not available to PlanT such as their actions and dynamics. Using this information, the expert determines the future position of all vehicles and estimates intersections between its own future position and those of the other vehicles to prevent most collisions. The dataset collected with this expert contains 228k frames. We use this as our reference point denoted by $1\times$. For our analysis, we also generate additional data following [26] but with different initializations of the traffic. The data quantities we use are always relative to the original dataset (i.e., $2\times$ contains double the data, $3\times$ contains triple). We refer the reader to [26] for a detailed description of the expert algorithm and dataset collection.

**Metrics.** We report the established metrics of the CARLA leaderboard [70]: *Route Completion (RC)*, *Infraction Score (IS)*, and *Driving Score (DS)*, which is the weighted average of the RC and IS. In addition, we show *Collisions with Vehicles per kilometer (CV)* and *Inference Time (IT)* for one forward pass of the model, measured in milliseconds on a single RTX 3080 GPU.

**Baselines.** To highlight the advantages of learning-based planning, we include a **rule-based** planning baseline that uses the same inputs as PlanT. It follows the same high-level algorithm as the expert but estimates the future of other vehicles using a constant speed assumption since it does not have access to their actions. **AIM-BEV** [18] is a recent privileged agent trained using IL. It uses a BEV semantic map input with channels for the road, lane markings, vehicles, and pedestrians, and a GRU identical to PlanT to predict a trajectory for the ego vehicle which is executed using lateral and longitudinal PID controllers. **Roach** [12] is a Reinforcement Learning (RL) based agent with a similar input representation as AIM-BEV that directly outputs driving actions. Roach and AIM-BEV are the closest existing methods to PlanT. However, they use a different input field of view in their representation leading to sub-optimal performance. We additionally build **PlanCNN**, a more competitive CNN-based approach for planning with the same training data and input information as PlanT, which is adapted from AIM-BEV to input a rasterized version of our object-level representation. We render the oriented vehicle bounding boxes in one channel, represent the speed of each pixel in a second channel, and render the oriented bounding boxes of the route in the third channel. We provide detailed descriptions of the baselines in the supplementary material.

**Implementation.** Our analysis includes three BERT encoder variants taken from [71]: MINI, SMALL, and MEDIUM with 11.2M, 28.8M and 41.4M parameters respectively. For PlanCNN, we experiment with two backbones: ResNet-18 and ResNet-34. We choose these architectures to maintain an IT which enables real-time execution. We train the models from scratch on 4 RTX 2080Ti GPUs with a total batch size of 128. Optimization is done with AdamW [72] for 47 epochs with an initial learning rate of $10^{-4}$ which we decay by 0.1 after 45 epochs. Training takes approximately 2.8, 3.4, and 4 hours for the three BERT variants on the $1\times$ dataset. We set the weight decay to 0.1 and clip the gradient norm at 1.0. For the auxiliary objective, we use quantization precisions $Z_a$ of 128 bins for the position, 4 bins for the speed and 32 bins for the orientation of the vehicles. We use $T_{in} = 0$ and $\delta t = 1$ for auxiliary supervision. The loss weight $\lambda$ is set to 0.2. By default, we use $D_{max} = 30\,\text{m}$, $N_s = 2$, and $L_{max} = 10\,\text{m}$. For our experiment with a full driving stack, we use a perception module based on TransFuser [26] to obtain the object-level input representation for PlanT. Additional details regarding this perception module as well as detailed ablation studies on the multi-task training and input representation hyperparameters are provided in the supplementary material.

| | Method | Input | DS ↑ | RC ↑ | IS ↑ | CV ↓ | IT ↓ |
|---|---|---|---|---|---|---|---|
| Sensor | LAV* [30] | Camera + LiDAR | 32.74±1.45 | 70.36±3.14 | 0.51±0.02 | **0.84±0.11** | - |
| | TransFuser [26] | Camera + LiDAR | 47.30±5.72 | **93.38±1.20** | 0.50±0.60 | 2.44±0.64 | 101.24 |
| | PlanT w/ perception | Camera + LiDAR | **57.66±5.01** | 88.20±0.94 | **0.65±0.06** | 0.97±0.09 | 37.61 |
| Privileged | Rule-based | Obj. + Route | 38.00±1.64 | 29.09±2.12 | 0.84±0.00 | 0.64±0.07 | - |
| | AIM-BEV [18] | Rast. Obj. + HD Map | 45.06±1.68 | 78.31±1.12 | 0.55±0.01 | 1.67±0.16 | 18.14 |
| | Roach* [12] | Rast. Obj. + HD Map | 55.27±1.43 | 88.16±1.52 | 0.62±0.02 | 0.76±0.07 | 3.24 |
| | PlanCNN | Rast. Obj. + Rast. Route | 77.47±1.34 | **94.53±2.59** | 0.81±0.03 | 0.43±0.05 | 28.94 |
| | PlanT | Obj. + Route | **81.36±6.54** | 93.55±2.62 | **0.87±0.05** | **0.31±0.12** | 10.79 |
| | *Expert [26]* | *Obj. + Route + Actions* | *76.91±2.23* | *88.67±0.56* | *0.86±0.03* | *0.28±0.06* | - |

Table 1: **Longest6 Results.** We show the mean±std for 3 evaluations. PlanT reaches expert-level performance and requires significantly less inference time than the baselines. *We evaluate the publicly available checkpoints at the time of submission for LAV and Roach. See supplementary material for results with checkpoints released later.

## 4.1 Obtaining Expert-Level Driving Performance

In the following, we discuss the key findings of our study which enable expert-level driving with learned planners. We begin with a discussion of the privileged methods and analyze the sensor-based methods in Section 4.2. Unless otherwise specified, the experiments consider the largest version of our dataset (3×) and models (MEDIUM for PlanT, ResNet-34 for PlanCNN).

**Input Representation.** Table 1 compares the performance on the Longest6 benchmark. The rule-based system acts cautiously and gets blocked often. Among the learning-based methods, both PlanCNN and PlanT significantly outperform AIM-BEV [18] and Roach [12]. We systematically break down the factors leading to this in Table 2a by studying the following: (1) the representation used for the road layout, (2) the horizontal field of view, (3) whether objects behind the ego vehicle are part of the representation, and (4) whether the input representation incorporates speed.

Roach uses the same view to the sides as AIM-BEV but additionally includes 8 m to the back and multiple input frames to reason about speed. We see in Table 2a that training PlanCNN in a configuration close to Roach (with the key differences being the removal of details from the map and a 0 m back view) results in a higher DS (59.97 vs. 55.27), demonstrating the importance of the route representation for urban driving. While additional information might be important when moving to more complex environments, our results suggest that the route is particularly important. Increasing the side view from 19.2 to 30 m improves PlanCNN from 59.97 to 70.72. Including vehicles to the rear further boosts PlanCNN's DS to 77.47 and improves PlanT's DS from 72.86 to 81.36. These results show that a full 360° field of view is helpful to handle certain situations encountered during our evaluation (e.g. changing lanes). Finally, removing the vehicle speed input significantly reduces the DS for both PlanCNN and PlanT (Table 2a), showing the importance of cues regarding motion.

**Scaling.** In Table 2b, we show the impact of scaling the dataset size and the model size for PlanT and PlanCNN. The circle size indicates the inference time (IT). First, we observe that PlanT demonstrates better data efficiency than PlanCNN, e.g., using the 1× data setting is sufficient to reach the same performance as PlanCNN with 2×. Interestingly, scaling the data from 1× to 3× leads to expert-level performance, showing the effectiveness of scaling. In fact, PlanT$_{MEDIUM}$ (81.36) outperforms the expert (76.91) in some evaluation runs. We visualize one consistent failure mode of the expert that leads to this discrepancy in Fig. 3a. We observe that the expert sometimes stops once it has already entered an intersection if it anticipates a collision, which then leads to collisions or blocked traffic. On the other hand, PlanT learns to wait further outside an intersection before entering which is a smoother function than the discrete rule-based expert and subsequently avoids these infractions. Importantly, in our final setting, PlanT$_{MEDIUM}$ is around 3× as fast as PlanCNN while being 4 points better in terms of the DS and PlanT$_{MINI}$ is 5.3× as fast (IT=5.46 ms) while reaching the same DS as PlanCNN. This shows that PlanT is suitable for systems where fast inference time is a requirement. We report results with multiple training seeds in the supplementary material.

**Loss.** A detailed study of the training strategy for PlanT can be found in the supplementary material, where we show that the auxiliary loss proposed in Eq. (4) is crucial to its performance. However, since this is a self-supervised objective, it can be incorporated without additional annotation costs. This is in line with recent findings on training transformers that show the effectiveness of supervising multiple output tokens instead of just a single [CLS] token [73].

| Method | Map | Side (m) | Back (m) | Speed | DS ↑ |
|---|---|---|---|---|---|
| AIM-BEV [18] | ✓ | 19.2 | 0 | - | 45.06±1.68 |
| Roach [12] | ✓ | 19.2 | 8 | ✓ | 55.27±1.43 |
| PlanCNN | - | 19.2 | 0 | ✓ | 59.97±4.47 |
|  | - | 30 | 0 | ✓ | 70.72±2.99 |
|  | - | 30 | 30 | ✓ | 77.47±1.34 |
|  | - | 30 | 30 | - | 69.13±1.43 |
| PlanT | - | 30 | 0 | ✓ | 72.86±5.56 |
|  | - | 30 | 30 | ✓ | **81.36±6.54** |
|  | - | 30 | 30 | - | 72.34±3.30 |

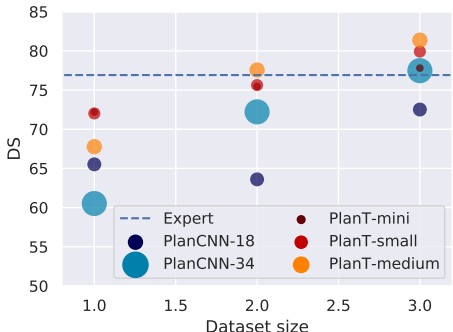

(a) **Input Representation.** DS on Longest6 (3 evaluation runs) with different input properties.

(b) **Scaling.** Mean DS on Longest6 for different dataset and model sizes. Circle size shows IT.

Table 2: We investigate the choices of the input representation (Table 2a) and architecture (Table 2b). for learning-based planners. Including vehicles to the back of the ego, encoding vehicle speeds, and scaling to large models/datasets is crucial for the performance of both PlanCNN and PlanT.

## 4.2 Combining an Off-the-Shelf Perception Module with PlanT

Next, we discuss the results of the sensor-based methods in Table 1. We compare the proposed approach to LAV [30] and TransFuser [26], which are recent state-of-the-art sensor-based methods. Our perception module is based on TransFuser, enabling a fair comparison to this approach. Therefore, PlanT with perception only detects vehicles to its front and has a limited view to the sides (16m instead of 30m). Our approach outperforms TransFuser [26] by 10.36 points and LAV by 24.92 points. While TransFuser uses an ensemble and manually designed heuristics to creep forward if stuck [26], these are unnecessary for PlanT with perception. Since we do not use an ensemble, we observe a $2.7\times$ speedup ($101.24\,\mathrm{ms}$ vs. $37.61\,\mathrm{ms}$) in IT compared to TransFuser. We refer to the supplementary material for a more detailed analysis.

## 4.3 Explainability: Identification of Most Relevant Objects

Finally, we investigate the explainability of PlanT and PlanCNN by analyzing the objects in the scene that are relevant and crucial for the agent's decision. In particular, we measure the relevance of an object in terms of the learned attention for PlanT and by considering the impact that the removal of each object has on the output predictions for PlanCNN. To quantify the ability to reason about the most relevant objects, we propose a novel evaluation scheme together with the *Relative Filtered Driving Score (RFDS)*. For the rule-based expert algorithm, collision avoidance depends on a single vehicle which it identifies as the reason for braking. To measure the RFDS of a learned planner, we run one forward pass of the planner (without executing the actions) to obtain a scalar *relevance score* for each vehicle in the scene. We then execute the expert algorithm while restricting its observations to the (single) vehicle with the highest relevance score. The RFDS is defined as the relative DS of this restricted version of the expert compared to the default version which checks for collisions against all vehicles. We describe the extraction of the relevance score for PlanT and PlanCNN in the following. Our protocol leads to a fair comparison of different agents as the RFDS does not depend on the ability to drive itself but only on the obtained ranking of object relevance.

**Baselines.** As a naïve baseline, we consider the inverse distance to the ego vehicle as a vehicle's relevance score, such that the expert only sees the closest vehicle. For PlanT, we extract the relevance score by adding the attention weights of all layers and heads for the [CLS] token. This only requires a single forward pass of PlanT. Since PlanCNN does not use attention, we choose a masking method to find the most salient region in the image, using the same principle as [59, 58, 57]. We remove one object at a time from the input image and compute the $L_1$ distance to the predicted waypoints for the full image. The objects are then ranked based on how much their absence affects the $L_1$ distance.

| Method | RFDS ↑ |
|---|---|
| Inverse Distance | 29.13±0.54 |
| PlanCNN + Masking | 82.83±6.79 |
| PlanT + Attention | **96.82±2.12** |

Table 3: **RFDS.** Relative score of the expert when only observing the most relevant vehicle according to the respective planner.

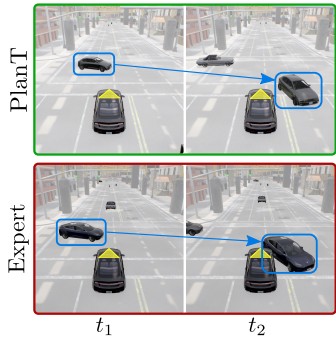

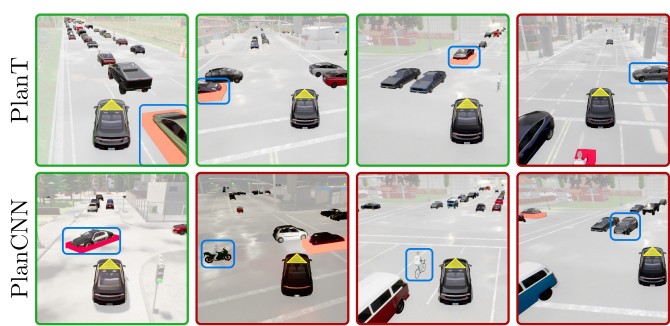

(a) **PlanT vs. Expert.** PlanT waits further outside the intersection than the expert to avoid a collision.

(b) **RFDS.** Vehicles with the highest relevance score are marked with a red bounding rectangle. We show examples for successful matching of relevance score and intuition (green frames) and failures (red frames).

Figure 3: We contrast a failure case of the expert to PlanT (Fig. 3a) and show the quality of the relevance scores (Fig. 3b). The ego vehicle is marked with a yellow triangle and vehicles that either lead to collisions or are intuitively the most relevant in the scene are marked with a blue box.

**Results.** We provide results for the reasoning about relevant objects in Table 3. Both planners significantly outperform the distance-based baseline, with PlanT obtaining a mean RFDS of 96.82 compared to 82.83 for PlanCNN. We show qualitative examples in Figure 3b where we highlight the vehicle with the highest relevance score using a red bounding rectangle. Both planners correctly identify the most important object in simple scenarios. However, PlanT is also able to correctly identify the most important object in complex scenes. When merging into a lane (examples 1 & 2 from the left) it correctly looks at the moving vehicles coming from the rear to avoid collisions. Example 3 shows advanced reasoning about dynamics. The two vehicles closer to the ego vehicle are moving away at a high speed and are therefore not as relevant. PlanT already pays attention to the more distant vehicle behind them as this is the one that it would collide with if it does not brake. One of the failures of PlanT we observe is that it sometimes allocates the highest attention to a very close vehicle behind itself (example 4) and misses the relevant object. PlanCNN has more prominent errors when there are a large number of vehicles in the scene or when merging lanes (examples 2 & 3). To better assess the driving performance and relevance scores we provide additional results in the supplementary video.

## 5 Conclusion

In this work, we take a step towards efficient, high-performance, explainable planning for autonomous driving with a novel object-level representation and transformer-based architecture called PlanT. Our experiments highlight the importance of correctly encoding the ego vehicle's route for planning. We show that incorporating a $360°$ field of view, information about vehicle speeds, and scaling up both the architecture and dataset size of a learned planner are essential to achieve state-of-the-art results. Additionally, PlanT significantly outperforms state-of-the-art end-to-end sensor-based models even with a noisy and incomplete input representation obtained via a perception module. Finally, we demonstrate that PlanT can reliably identify the most relevant object in the scene via a new metric and evaluation protocol that measure explainability.

**Limitations.** Firstly, the expert driver used in our IL-based training strategy does not achieve a perfect score (Table 1) and has certain consistent failure modes (Fig. 3a, more examples in the supplementary material). Human data collection to address this would be time-consuming (the $3\times$ dataset used in our experiments contains around 95 hours of driving). Second, all our experiments are conducted in simulation. Real-world scenarios are more diverse and challenging. However, CARLA is a high-fidelity simulator actively used by many researchers for autonomous driving, and previous findings demonstrate that systems developed in simulators like CARLA can be transferred to the real world [31, 74, 75]. Finally, our experiment with perception (Section 4.2) uses a single off-the-shelf perception module that was not specifically optimized for PlanT, leading to sub-optimal performance. It is a well-known limitation of modular systems that downstream modules cannot recover easily from errors made by earlier modules. A thorough analysis of perception robustness and uncertainty encoding are important research directions beyond the scope of this work.

**Acknowledgments**

This work was supported by the BMWi (KI Delta Learning, project number: 19A19013O), the BMBF (Tübingen AI Center, FKZ: 01IS18039A), the DFG (SFB 1233, TP 17, project number: 276693517), by the ERC (853489 - DEXIM), and by EXC (number 2064/1 – project number 390727645). We thank the International Max Planck Research School for Intelligent Systems (IMPRS-IS) for supporting K. Renz, K. Chitta and O.-B. Mercea. The authors also thank Niklas Hanselmann and Markus Flicke for proofreading and Bernhard Jaeger for helpful discussions.

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
