# OpenReview forum: "PlanT: Explainable Planning Transformers via Object-Level Representations"
_robot-learning.org/CoRL/2022/Conference — CoRL 2022 Poster_

### Official Review · Reviewer_H8PS · 2022-07-20

**Originality:** Good
**Technical Quality:** Very Good
**Clarity Of Presentation:** Good
**Impact:** 3

**Recommendation:**

Weak Accept: I recommend accepting the paper, but will not argue for my recommendation if the majority of other reviewers have a different opinion.

**Summary:**

This paper presents a transformer-based approach for planning (PlanT) in the context of autonomous driving that takes as input a sparse object-based representation of the environment (in contrast to the denser HD maps commonly used in practice). The approach is demonstrated in challenging simulated autonomous driving scenarios and surpasses prior methods (many based on denser input representations) across several benchmark driving metrics.

**Issues:**

See Weaknesses (reproduced here)

- The technical exposition in Sec 3 is very dense. I believe this is in part due to the fact that essentially the entire system is described verbally. I could see this making it difficult for a practitioner to attempt to reproduce this method without access to the code. I would encourage the authors to try to define concisely (even at a high-level) the mathematical model captured in part by Figure 1. In particular, with the current presentation, it is tremendously difficult to visualize how the verbally described components fit together (even with the aid of Fig. 2), or determine any sort of relative importance (i.e. what are really the "main" ideas versus auxiliary components). It seems that almost everything is essentially given the same "priority" in the current presentation. In the absence of such a mathematical model, it would help to give an overview figure, perhaps one similar to Fig. 2, but to provide references to the sections in text where components are discussed. The pair of eq (1) with Fig 2 is a great example of this working successfully. I particularly appreciated that the different loss terms were cross references, and the choice of notation here helped convey that. I wonder if a similar approach could be taken for the rest of the system?

- Strictly speaking, is it fair to call the output of the system "explainable"? I am not an expert in explainable AI, but while the object attention weights visualized in the supplemental video for PlanT do look more "stable" than PlanCNN, it was rarely obvious to me why PlanT highlighted particular car as "relevant" (except in the event of the obvious interpretation that the most relevant car was simply the one immediately in front of the agent). Independent of this, the ability to provide these attention weights seems interesting to me, and does (I believe) help visualize or interpret the internals of the model, compared to a black-box model.

- The downside of the sparseness of the object-based representation in practice is that it may be more error prone than an HD map, for the simple reason that consistency of the (denser) information encoded in the HD map can be used to reason about e.g. outliers. Another natural issue is that the object-based map will typically rely crucially on the output of learned perception models, which can produce spurious detections or misclassifications. The limitations section already mentions that object states were extracted from the simulator directly (and it is reasonable in this work to focus on planning specifically), however, I would suggest discussing the issue of robustness, particularly with respect to spurious or missed object detections and misclassifications. In principle, this may be something that could be tested in simulation.

- Minor, but one of the stated contributions is that "we demonstrate that a simple object-level representation is sufficient to encode all the information relevant for planning in urban driving environments." This seems like far too broad a claim to be made without more thorough evaluation in real environments.

**Quality Of The Limitations Section:**

Limitations are addressed clearly

**Reviewer Expertise:**

3: The reviewer is fairly confident that the evaluation is correct

**Robotics Focus:**

Relevant but unlikely to deploy to hardware in near future

**Strengths And Weaknesses:**

# Strengths

- The technical approach, as I understand it, seems sound. The idea of using a purely object-based representation as input to a learned planner seems neat, and the ability to assign some scores to objects related to their "relevance" seems interesting.

- The fact that the planner here takes a *sparse* representation is particularly interesting, as this requires significantly less data than HD maps currently in use (making it also potentially easier to process at high rates).

- The supplemental video was **very** helpful in understanding the system and visualizing the output.

- The experimental evaluation is very thorough, and the detailed appendices are appreciated.

# Weaknesses
(I am combining weaknesses here with issues/clarifications)

- The technical exposition in Sec 3 is very dense. I believe this is in part due to the fact that essentially the entire system is described verbally. I could see this making it difficult for a practitioner to attempt to reproduce this method without access to the code. I would encourage the authors to try to define concisely (even at a high-level) the mathematical model captured in part by Figure 1. In particular, with the current presentation, it is tremendously difficult to visualize how the verbally described components fit together (even with the aid of Fig. 2), or determine any sort of relative importance (i.e. what are really the "main" ideas versus auxiliary components). It seems that almost everything is essentially given the same "priority" in the current presentation. In the absence of such a mathematical model, it would help to give an overview figure, perhaps one similar to Fig. 2, but to provide references to the sections in text where components are discussed. The pair of eq (1) with Fig 2 is a great example of this working successfully. I particularly appreciated that the different loss terms were cross references, and the choice of notation here helped convey that. I wonder if a similar approach could be taken for the rest of the system?

- Strictly speaking, is it fair to call the output of the system "explainable"? I am not an expert in explainable AI, but while the object attention weights visualized in the supplemental video for PlanT do look more "stable" than PlanCNN, it was rarely obvious to me why PlanT highlighted particular car as "relevant" (except in the event of the obvious interpretation that the most relevant car was simply the one immediately in front of the agent). Independent of this, the ability to provide these attention weights seems interesting to me, and does (I believe) help visualize or interpret the internals of the model, compared to a black-box model.

- The downside of the sparseness of the object-based representation in practice is that it may be more error prone than an HD map, for the simple reason that consistency of the (denser) information encoded in the HD map can be used to reason about e.g. outliers. Another natural issue is that the object-based map will typically rely crucially on the output of learned perception models, which can produce spurious detections or misclassifications. The limitations section already mentions that object states were extracted from the simulator directly (and it is reasonable in this work to focus on planning specifically), however, I would suggest discussing the issue of robustness, particularly with respect to spurious or missed object detections and misclassifications. In principle, this may be something that could be tested in simulation.

- Minor, but one of the stated contributions is that "we demonstrate that a simple object-level representation is sufficient to encode all the information relevant for planning in urban driving environments." This seems like far too broad a claim to be made without more thorough evaluation in real environments.

**Summary Of Recommendation:**

The contribution here (a transformer-based planning approach that processes a parsimonious object-based representation for autonomous driving) seems relevant and novel to me. The paper overall is good, but there are some clarity issues with the technical presentation; this is where it could be improved the most, in my opinion. The evaluation is thorough and clear.

---

> ### Author Response · Authors · 2022-08-25
> **Response to Reviewer H8PS**
>
> Thank you for your time and feedback to help improve the quality of our paper.
>
> &nbsp;
>
> 1. **Writing:** Thanks for this great suggestion. We will include your suggestions in our revised submission (which will be uploaded shortly).
>
> &nbsp;
>
> 2. **Explainability:** Since the judgment from only qualitative examples of whether the planner attends to the “important” object is subjective, we introduced the metric RFDS (Table 3) measuring the correctness of the attention. We show quantitatively that PlanT outperforms the CNN-based method by 14%. We agree that the attention scores do not directly give an explanation of why the model attended to a certain object. However, as also mentioned in the review, they do help interpret the decisions of the vehicle. Our visualizations could also be used to increase trustworthiness when displayed to the passengers of a self-driving vehicle.
>
> &nbsp;
>
> 3. **Sparseness and robustness:**
>
>      **(1) Sparseness:** Removing the need for an HD map has several advantages [1]. Primarily, it is very expensive and time-consuming to create and maintain HD maps. Since our PlanT architecture can be theoretically extended by adding more tokens for different object types, encoding other parts of the map is possible and an interesting future direction when more complex benchmarks are available. For a discussion about our claim about the sufficiency of the route, please refer to our [general response to all reviewers](https://openreview.net/forum?id=80vpxjt3vq&noteId=MgulWWjQJF-).
>
>      **(2) Robustness:** It is indeed a well-known limitation of modular systems that downstream modules cannot recover easily from errors made by earlier modules. A thorough analysis of the level of robustness to different perception systems is an interesting direction but was not feasible in the limited time frame of the rebuttal phase. However, we add a perception experiment with one specific perception system to show that even with some spurious or missed detections we can outperform current state-of-the-art models that are trained end-to-end. We refer to our [general response to all reviewers](https://openreview.net/forum?id=80vpxjt3vq&noteId=MgulWWjQJF-) for the exact results of the experiment. We will mention robustness with respect to spurious or missed object detections and misclassifications in the section discussing our perception experiment.
>
> &nbsp;
>
> 4. **Claim about sparse representation:** We will update the corresponding claim in our revised submission (which will be uploaded shortly) to clarify this point.
>
> &nbsp;
>
> ### Reference
> [1] Sergio Casas, Abbas Sadat and Raquel Urtasun. “MP3: A Unified Model to Map, Perceive, Predict and Plan.” In CVPR, 2021.

---

### Official Review · Reviewer_LmXW · 2022-07-29

**Originality:** Excellent
**Technical Quality:** Very Good
**Clarity Of Presentation:** Very Good
**Impact:** 4

**Recommendation:**

Strong Accept: I recommend accepting the paper and will argue for my recommendation even if other reviewers hold a different opinion.

**Summary:**

This paper designs a novel object-level scene representation and proposes a transformer based framework to predict the set of waypoints given the route. The framework outperforms other baselines and is more efficient. The attention mechanism also improves the explain explainability of the method.

**Issues:**

Some of the training details like the illustration of the input representations and training strategy should be replaced to the body part.

**Quality Of The Limitations Section:**

Limitations are addressed clearly

**Reviewer Expertise:**

4: The reviewer is confident but not absolutely certain that the evaluation is correct

**Robotics Focus:**

Highly relevant to robotics but no hardware experiments

**Strengths And Weaknesses:**

Strengths:
1.Transform the scene representation from pixel level to object level which is closer to human perception.
2.Regarding the vehicles and routes as a sequence and utilize transformer to predict the waypoints, which is a novel and reasonable combination.
3.The visualization of the attention score over other vehicles improves the explainability of the decision making process.
4.The attached videos clearly demonstrates the effectiveness of the framework.

Weaknesses:
The framework only considers the controlling process but ignore the perception process, which greatly simplifies the problem. I hope that the future work would take perception into consideration and construct a complete framework.

**Summary Of Recommendation:**

This paper designs a novel object-level scene representation and proposes a transformer based framework to predict the set of waypoints given the route. The authors transform the scene representation from pixel level to object level which is closer to human perception by regarding the vehicles and routes as a sequence and utilize transformer to predict the waypoints, which is a novel and reasonable combination.  The framework outperforms other baselines and is more efficient. The attention mechanism also improves the explain explainability of the method. However, the framework only considers the controlling process but ignore the perception process, which greatly simplifies the problem.

---

> ### Author Response · Authors · 2022-08-25
> **Response to Reviewer LmXW**
>
> Thank you for your time and feedback to help improve the quality of our paper. We appreciate that the review nicely highlights the key strengths of our paper: our system outperforms other baselines, is more efficient, and its attention mechanism improves explainability.
>
>
> **Perception:**  We refer to our [general response to all reviewers](https://openreview.net/forum?id=80vpxjt3vq&noteId=BAK_iA8SMb9) where we show that we can reach state-of-the-art performance on the Longest6 Benchmark and add a detailed discussion justifying our assumption.

---

### Official Review · Reviewer_QCRj · 2022-07-29

**Originality:** Good
**Technical Quality:** Very Good
**Clarity Of Presentation:** Excellent
**Impact:** 3

**Recommendation:**

Weak Accept: I recommend accepting the paper, but will not argue for my recommendation if the majority of other reviewers have a different opinion.

**Summary:**

This paper presents PlanT, a transformer architecture for motion planning. Unlike most models which reason according to rasterized BEV scene representations, PlanT leverages an object-centric input representation and simple route encoding. The model is trained similar to prior works using behavior cloning with supervision obtained from a rules-based expert with access to privileged information. The authors claim PlanT's architecture and input representation leads to improved planning performance and more explainable outputs, since the transformer's attention weights provide a measure of importance of each object in the scene. The authors provide a set of comprehensive experiments in the CARLA simulator to support these claims.

**Issues:**

See weaknesses section.

Some additional minor comments:
- I personally find the title of this paper misleading. I think it would be more accurate to include "transformer" and "object-centric" as those highlight the key contributions of the paper.
- Re the auxiliary prediction loss the authors say, "However, since this is a self-supervised objective, it can be incorporated without additional annotation costs". I find this statement misleading as it is only true in the simulated setting and may not necessarily be true in a real-world setting. It's not clear for real-world data whether one would use the outputs from the perception system (which would be noisy, but not require annotation) or GT human labels (which would be more accurate, but costly to annotate).

**Quality Of The Limitations Section:**

Limitations are addressed clearly

**Reviewer Expertise:**

3: The reviewer is fairly confident that the evaluation is correct

**Robotics Focus:**

Relevant but unlikely to deploy to hardware in near future

**Strengths And Weaknesses:**

**Strengths**
- **Clear presentation:** The paper is very well written and easy to follow.
- **Very thorough experiments:** I really appreciate the scientific rigour in the experimental analysis and how the authors control for each change between the models. I appreciate the authors comparing against PlanCNN and found the analysis in Table (2a) very well done as it shows exactly where the performance improvements of the model come from.
- **Simple yet effective model:** The proposed model is quite simple, yet is convincingly shown to be the best performing model. The qualitative difference between the explainability visualization of PlanCNN and PlanT shown in the video is apparent and helps support the claim that PlanT is a more explainable model.

**Weaknesses**
- **Assumes perfect perception**: As mentioned in the limitations, this paper only considers the problem of planning and assumes perfect perception. While I understand the desire to reduce the scope of the problem, I believe the experimental section or supplementary should show at least one experiment with end-to-end results (from sensor inputs). This would help provide a sense of what the gap is between using GT inputs vs. noisy perception.
- **Simplified route representation:** While it was an interesting contribution to the paper that a simplified route is sufficient for state-of-the-art performance, this seems like a limitation for a real-world setting. How would this representation support a discretionary lane change (one that isn't needed to follow a route, but that the planner performs for a safer and/or faster driving)? What is the intuition for why all lane information is not helpful for better prediction of the other actors?
- **Missing connection to transformers for motion-forecasting:** While the end objective is clearly different, the problems of motion forecasting and motion planning share many similarities. Therefore, I think it could be beneficial to mention the connections to the many recent works on leveraging transformers for motion-forecasting and highlight the differences in this architecture. I find that the proposed architecture has similarities with SceneTransformer [1], which also leverages a Transformer to predict the future states of each actor - I believe the key differences between the two are the end objective, the architectural details (cross-attention) and the route representation used in this paper. Additionally, past works in transformers for motion-forecasting [2] have also visualized attention weights to understand the significance actors have on each other (albeit for the motion forecasting task), which may be worth mentioning in the related works.
- **Only tested in simulation and reliance on a rules-based expert:** This was also mentioned in the paper's limitations, but I'm curious how the authors would imagine using this system in the real-world? Since the model is data hungry, it seems like it would be most feasible to train in simulation and try to transfer the model to the real-world? If that is the case, why leverage an expert and not consider other training schemes like RL? When using the same field of view, it seems like Roach and PlanCNN are within performance (accounting for the confidence intervals). Do you think this model could be trained or fine-tuned with RL in the future to outperform the expert even further?

References:
- [1] Ngiam et al., Scene Transformer: A unified architecture for predicting multiple agent trajectories, ICLR 2022
- [2] Li et al., End-to-end Contextual Perception and Prediction with Interaction Transformer, IROS 2020

**Summary Of Recommendation:**

I am on the fence for this paper but leaning towards accept. On the one hand, I think this method has limited novelty (when considered in the context of recent transformers for motion forecasting) and has several limitations (most of which are  mentioned in the paper's limitations section) which would need to be addressed before it could be deployed in a real-world setting. However, many of these simplifying assumptions are common in the research community and I find the experimental analysis of this paper very thorough and the detailed ablations and experiments insightful. Additionally, while there are similarities with prior work, applying the transformer to object-centric representations for motion planning is novel, as far as I know. Therefore, since the technical quality is high, the experiments are insightful and the contributions could provide value to the community, I would lean towards acceptance.

---

> ### Author Response · Authors · 2022-08-25
> **Response to Reviewer QCRj**
>
> Thank you for your time and feedback to help improve the quality of our paper.
>
> &nbsp;
>
> 1. **Perception:**  We refer to our [general response to all reviewers](https://openreview.net/forum?id=80vpxjt3vq&noteId=BAK_iA8SMb9) where we show that we can reach state-of-the-art performance on the Longest6 Benchmark and add a detailed discussion justifying our assumption.
>
> &nbsp;
>
> 2. **Simplified route representation:** We refer to our [general response to all reviewers](https://openreview.net/forum?id=80vpxjt3vq&noteId=MgulWWjQJF-) about a discussion of the sufficiency of the route representation.
>
> &nbsp;
>
> 3. **Motion-Forecasting:** Thanks for pointing this out. We will add the missing references and a discussion to our related work section. We agree with the reviewer’s comment regarding the key differences (the end objective, the architectural details and the route representation) and would like to highlight in particular the difference in metrics and evaluation schemes. It has been shown that offline metrics used by existing motion forecasting papers do not correlate well with closed-loop driving performance [1].
>
> &nbsp;
>
> 4. **Real-world usage and reliance on rule-based experts:**
>
>      **(1) Real-world usage:** As Imitation Learning approaches do not need direct interaction with the environment during training, PlanT can also be trained with real-world data. Real-world driving datasets such as NuPlan [2] can be used to train the model.
>
>      **(2) Using Reinforcement Learning (RL)** to train or fine-tune the model is feasible and indeed an interesting direction to surpass the performance of a rule-based expert. However, training with RL is more challenging and current state-of-the-art methods use Imitation Learning. In particular, RL is less sample-efficient, harder to scale, and requires carefully designed reward functions.
>
>
> &nbsp;
>
>
>
> ### Response to minor comments:
> - **Title:** We will change the title to: “PlanT: Explainable Planning Transformers via Object-Level Representations”
>
>
> - **Auxiliary prediction loss:** Our loss function involves 2 terms: waypoint prediction (main task) and vehicle future prediction (auxiliary task). We would like to clarify that our statement (“without additional annotation costs”) refers to the fact that including the auxiliary task does not require any new labeling costs compared to training with only the main task i.e., it is self-supervised. This is due to the fact that in order to train the network to perform our main task, we already require the labels for attributes of the other vehicles, as this is part of our input representation. To obtain these attributes for real-world data, some existing datasets [2, 3] use offline perception systems to perform auto-labeling. This leads to higher quality labels compared to an on-board perception system and circumvents the need to manually annotate the data. For the auxiliary prediction loss, we would simply retrieve vehicle attribute labels from the next time step.
> As this was not clearly stated in the submitted version of the paper we will clarify it in the revised submission (which will be uploaded shortly).
>
> &nbsp;
>
> ### References
> [1] Felipe Codevilla, Antonio M. López, Vladlen Koltun and Alexey Dosovitskiy. “On Offline Evaluation of Vision-based Driving Models.” In ECCV, 2018.
>
> [2] Holger Caesar, Juraj Kabzan, Kok Seang Tan, Whye Kit Fong, Eric M. Wolff, Alex Lang, Luke Fletcher, Oscar Beijbom and Sammy Omari. “nuPlan: A closed-loop ML-based planning benchmark for autonomous vehicles.” In ArXiv, 2021.
>
> [3] Scott M. Ettinger, Shuyang Cheng, Benjamin Caine, Chenxi Liu, Hang Zhao, Sabeek Pradhan, Yuning Chai, Benjamin Sapp, C. Qi, Yin Zhou, Zoey Yang, Aurelien Chouard, Pei Sun, Jiquan Ngiam, Vijay Vasudevan, Alexander McCauley, Jonathon Shlens and Drago Anguelov. “Large Scale Interactive Motion Forecasting for Autonomous Driving: The Waymo Open Motion Dataset.” In ICCV, 2021.

---

### Official Review · Reviewer_ZSWX · 2022-07-31

**Originality:** Poor
**Technical Quality:** Fair
**Clarity Of Presentation:** Very Good
**Impact:** 3

**Recommendation:**

Strong Reject: I recommend rejecting the paper and will argue for my recommendation even if other reviewers hold a different opinion.

**Summary:**

The paper proposes PlanT, a transformer-based end-to-end planner that uses object-level representations. The idea is very simple, the transformer takes vehicle bounding boxes as well as a discretized route as input tokens and outputs the planned future waypoints. Additionally, they predict the future of other vehicles in the scene as an auxiliary task that is shown to be helpful for better representation learning. The contributions are the method itself as well as a metric to measure the explainability of the planner.

**Issues:**

Please see weaknesses section. I recommend that the authors add a perception module, repeat their experiments, revisit the claims and resubmit to a subsequent conference. While I believe the work is interesting and it is worth continue building on top of, with the experiments as they stand I don't believe readers can take any valuable conclusion.

**Quality Of The Limitations Section:**

Limitations are not well addressed

**Reviewer Expertise:**

5: The reviewer is absolutely certain that the evaluation is correct and very familiar with the relevant literature

**Robotics Focus:**

Relevant but unlikely to deploy to hardware in near future

**Strengths And Weaknesses:**

### Strengths
- Paper is written clearly
- The motivation for sparse representations outlined in the abstract is fair.
- Method is simple and easy to understand
- The proposed metric to evaluate the explainability of a planner, RFDS, is interesting. However, I believe the usefulness of the metric to drive progress in the field is questionable, as it only gets at the attention placed on other objects.

### Weaknesses
- **Main weakness:** the paper claims as a contribution that they demonstrate that "a simple object-level representation is sufficient to encode all the information relevant for planning in urban driving environments". This claim is false, and the reason is that the proposed method assumes perfect perception. While this limitation is recognized in the limitations section, its importance is well beyond what the paper assumes. The reason why other papers use BEV dense representations is to *capture uncertainty*. When not having perfect perception and dealing with challenges such as occlusion, uncertainty on the current and future states of objects grows very significantly. Since the proposed method accesses the internal state of the simulator to know the object locations, speed, etc., there is no uncertainty in the current states of objects, and very little on future ones since CARLA's actors are overly simplistic. Since there is no uncertainty about object existence, all the information can be captured at the object level and thus the proposed method work best. However, this is not a realistic robotic application. Even if the authors would do experiments with a real object detector, CARLA is not the most suitable environment for that as perception is much easier than in the real-world.
- The paper also claims to be 5.3x faster than previous method. First, that is not obvious from Table 1. Moreover, even if the claim were true with the current experiments, we come back to the fact that the method doesn't perform object detection. Running object detection would impact inference time in two ways. First, the time to run the object detector. Second, the number of actors would increase significantly to attain an acceptable recall, further increasing runtime.
- It is also claimed that route is all the map representation needed. Again, I believe this is true for their experiments, but an flawed argument in reality. First, CARLA's scenarios are very simple and the other actors quite reactive. This makes it easy for the SDV to never have to steer off the route to perform mitigation maneuvers like a safe stop. Second, we come back to assuming perfect perception. Since the method has a privileged understanding of the scene there is much less uncertainty, which translates into the SDV never having to steer off the route.
- While the attention from SDV token to other actor token is interesting as an interpretable proxy for importance, this isn't really novel and was previously proposed by CAPO (see citation below).
- It is not fully clear what is the output representation for the auxiliary task of predicting the future trajectories of other agents. As a consequence, $\mathcal{L}_{CE}$ is also unclear.
```
@inproceedings{mcallister2022control,
  title={Control-Aware Prediction Objectives for Autonomous Driving},
  author={Rowan McAllister and Blake Wulfe and Jean Mercat and Logan Ellis and Sergey Levine and Adrien Gaidon},
  booktitle={International Conference on Robotics and Automation (ICRA)},
  year={2022}
}
```

**Summary Of Recommendation:**

I recommend rejection for this paper given that the claims are not validated by the method and experimental setup.

---

> ### Author Response · Authors · 2022-08-25
> **References for Response to Reviewer ZSWX**
>
> [1] Holger Caesar, Varun Bankiti, Alex H. Lang, Sourabh Vora, Venice Erin Liong, Qiang Xu, Anush Krishnan, Yu Pan, Giancarlo Baldan, and Oscar Beijbom. nuscenes: A multimodal dataset for autonomous driving. In CVPR, 2020
>
> [2] Ming-Fang Chang, John W Lambert, Patsorn Sangkloy, Jagjeet Singh, Slawomir Bak, Andrew Hartnett, De Wang, Peter Carr, Simon Lucey, Deva Ramanan, and James Hays. Argoverse: 3d tracking and forecasting with rich maps. In CVPR, 2019.
>
> [3] Scott M. Ettinger, Shuyang Cheng, Benjamin Caine, Chenxi Liu, Hang Zhao, Sabeek Pradhan, Yuning Chai, Benjamin Sapp, C. Qi, Yin Zhou, Zoey Yang, Aurelien Chouard, Pei Sun, Jiquan Ngiam, Vijay Vasudevan, Alexander McCauley, Jonathon Shlens and Drago Anguelov. “Large Scale Interactive Motion Forecasting for Autonomous Driving : The Waymo Open Motion Dataset.” In ICCV, 2021.
>
> [4] Johnny L. Houston, Guido C. A. Zuidhof, Luca Bergamini, Yawei Ye, Ashesh Jain, Sammy Omari, Vladimir I. Iglovikov and Peter Ondruska. “One Thousand and One Hours: Self-driving Motion Prediction Dataset.” In CoRL, 2020.
>
> [5] Holger Caesar, Juraj Kabzan, Kok Seang Tan, Whye Kit Fong, Eric M. Wolff, Alex Lang, Luke Fletcher, Oscar Beijbom and Sammy Omari. “nuPlan: A closed-loop ML-based planning benchmark for autonomous vehicles.” In ArXiv, 2021.
>
> [6] Marin Toromanoff, Émilie Wirbel and Fabien Moutarde. “End-to-End Model-Free Reinforcement Learning for Urban Driving Using Implicit Affordances.” In CVPR, 2020.
>
> [7] Kashyap Chitta, Aditya Prakash and Andreas Geiger. “NEAT: Neural Attention Fields for End-to-End Autonomous Driving.” In ICCV, 2021.
>
> [8] Dian Chen, Vladlen Koltun and Philipp Krähenbühl. "Learning to drive from a world on rails," In ICCV, 2021.
>
> [9] Zhejun Zhang, Alexander Liniger, Dengxin Dai, Fisher Yu and Luc Van Gool. “End-to-End Urban Driving by Imitating a Reinforcement Learning Coach.” In ICCV, 2021.
>
> [10] Jimuyang Zhang and Eshed Ohn-Bar. “Learning by Watching.” In CVPR, 2021.
>
> [11] Rowan McAllister, Blake Wulfe, Jean Mercat, Logan Ellis, Sergey Levine, and Adrien Gaidon.”Control-Aware Prediction Objectives for Autonomous Driving,” In ICRA, 2022.
>
> [12] Niklas Hanselmann, Katrin Renz, Kashyap Chitta, Apratim Bhattacharyya and Andreas Geiger. “KING: Generating Safety-Critical Driving Scenarios for Robust Imitation via Kinematics Gradients.” In ECCV, 2022.
>
> [13] Kashyap Chitta, Aditya Prakash, Bernhard Jaeger, Zehao Yu, Katrin Renz and Andreas Geiger. “TransFuser: Imitation with Transformer-Based Sensor Fusion for Autonomous Driving.” In TPAMI, 2022.
>
> [14] Felipe Codevilla, Antonio M. López, Vladlen Koltun and Alexey Dosovitskiy. “On Offline Evaluation of Vision-based Driving Models.” In ECCV, 2018.

---

> ### Author Response · Authors · 2022-08-25
> **Response to Reviewer ZSWX**
>
> Thank you for your time and feedback to help improve the quality of our paper.
> &nbsp;
>
> &nbsp;
>
> 1. **Perfect perception + simplicity of CARLA**
>
>      **(1) Perfect perception:** We refer to our [general response to all reviewers](https://openreview.net/forum?id=80vpxjt3vq&noteId=BAK_iA8SMb9) where we show that we can reach state-of-the-art performance on the Longest6 benchmark and add a detailed discussion justifying our assumption of perfect perception. Furthermore, we will mention the issue of uncertainty handling in our revised submission. Specifically, we would like to point out that it is straightforward to associate a scalar uncertainty value with any attribute of each object in the proposed representation if this is required for more challenging planning benchmarks in the future.
>
>      **(2) Simplicity of CARLA:** We agree that CARLA cannot capture the full complexity of the real world. However, existing real-world datasets and benchmarks focus on perception or motion forecasting [1, 2, 3, 4] and do not allow closed-loop testing. It has been shown that offline metrics used by existing real-world datasets do not correlate well with closed-loop driving performance [14]. Therefore, even though the traffic and perception are more realistic, these datasets are not suitable for evaluating our model. Due to the complexity of closed-loop testing, there is no real-world closed-loop planning benchmark which could be used as an alternative for our experiments. CARLA is an established simulation environment which is actively used by many researchers for autonomous driving [6, 7, 8, 9, 10, 11, 12, 13]. Importantly, CARLA is a sufficiently challenging environment with regards to both planning and perception, as evidenced by methods on the [CARLA Leaderboard](https://leaderboard.carla.org/leaderboard/) which are still unable to solve the existing scenarios. We believe that our analysis is a valuable step forward for the community given the current state of the art, and that the success of PlanT in simulation is an important prerequisite for success in more challenging real-world scenarios.
>
> &nbsp;
>
> 2. **Inference time:**
>
>       **(1) Highlighting inference time results:** It is true that the result for the 5.3x improvement in inference speed is not obvious from Table 1. This factor is taken from Section 4.1 (subsection: PlanT vs. PlanCNN) where we explicitly mention that “PlanT-mini is 5.3× as fast (IT=5.46 ms) while reaching the same DS as PlanCNN.“  We will further highlight the inference time results in the revised submission through a dedicated section in the supplementary material, and include the discussion below regarding runtime for a full driving system that includes a perception module in the main draft.
>
>       **(2) Runtime with an added perception module:** It is true that for calculating the runtime of the full driving stack the inference time of the object detector needs to be added to the inference time of the planner. However, we would like to emphasize the importance of improving the inference time of the planner, which is specifically what we claim in the paper. When considering real-time requirements of the full driving stack (around 30 fps, i.e., 33 ms), decreasing the inference time of the planner to 5.46 ms compared to 28.94 ms without any loss in performance would permit the use of a significantly more accurate object detector while maintaining the required system runtime. Furthermore, we agree that an increase in the number of input actors for PlanT may occur in environments where the perception task is more challenging. To evaluate the impact of such an increase, we scale the number of input vehicle tokens to 2x the true number of vehicles in every frame, and observe that the planner runtime increases by only 9.6%. Therefore, our conclusions regarding the improvements in inference time for PlanT compared to PlanCNN remain true in a setting involving false positive detections.
>
> &nbsp;
>
> 3. **Route vs. Map:**
>      We refer to our [general response to all reviewers](https://openreview.net/forum?id=80vpxjt3vq&noteId=MgulWWjQJF-) about a discussion of the sufficiency of the route representation.
>
> &nbsp;
>
> 4. **Attention Scores:**
>      Thanks for pointing us to this paper. We will add a discussion about this concurrent work in the revised version of the paper. CAPO indeed also uses attention scores but acts in a far simpler setting where they only consider pedestrian-ego interactions on a short straight route. In contrast, we show that the attention scores are meaningful when evaluating the full planning task in an urban environment with dense traffic.
>
> &nbsp;
>
> 5. **Auxiliary Task:** We will improve the clarity of this section and upload a revised submission.

---

### Author Response · Authors · 2022-08-25
**General response to all reviewers (1/2)**

We thank all the reviewers and the AC for their time and valuable feedback. We are happy that the reviewers appreciate our “simple yet effective model” [[QCRj](https://openreview.net/forum?id=80vpxjt3vq&noteId=k4XIJDyPWH1)] which is “easy to understand” [[ZSWX](https://openreview.net/forum?id=80vpxjt3vq&noteId=Iq5Anjl-v6Z)] and whose attention mechanism “improves the explainability of the decision making” [[LmXW](https://openreview.net/forum?id=80vpxjt3vq&noteId=kBRZSe15Vls7)]. Furthermore, the reviewers highlighted our “thorough” experimental evaluation [[H8PS](https://openreview.net/forum?id=80vpxjt3vq&noteId=NMkOfKv4B_Z)] and “scientific rigour in the experimental analysis” [[QCRj](https://openreview.net/forum?id=80vpxjt3vq&noteId=k4XIJDyPWH1)], as well as the fact that the “paper is written clearly” [[ZSWX](https://openreview.net/forum?id=80vpxjt3vq&noteId=Iq5Anjl-v6Z)] and “easy to follow” [[QCRj](https://openreview.net/forum?id=80vpxjt3vq&noteId=k4XIJDyPWH1)].

We appreciate the reviewers’ helpful suggestions to strengthen our work and we will upload a revision of our submission which incorporates these.

In this comment, we justify the assumption of perfect perception for our proposed model which was mentioned by all reviewers. We show state-of-the-art results on the Longest6 benchmark for PlanT when we incorporate a perception module. Additionally, we discuss the claim about the sufficiency of the route in our representation. We respond to all other comments and concerns of the reviewers and the AC separately below.

## Assumption of perfect perception
In our work, we focus on investigating planning in isolation to identify the most critical elements for planning and not entangle our conclusions with factors influenced by perception. [[H8PS](https://openreview.net/forum?id=80vpxjt3vq&noteId=NMkOfKv4B_Z)] also stated in their review that it is “reasonable in this work to focus on planning specifically”, and [[QCRj](https://openreview.net/forum?id=80vpxjt3vq&noteId=k4XIJDyPWH1)] noticed that “these simplifying assumptions are common in the research community”. There were several reasons that led us to this choice of experimental setting:

**(1) Common assumption in related work**: Using ground truth information about other objects is a common approach in planning [1,2,3,4]. CAPO [5] (ICRA’22) uses ground-truth trajectories of agents in their approach. KING [6] (ECCV’22) and ROACH [7] (ICCV’21) present several experiments with ground truth information taken from the CARLA simulator. The concurrent work DriveIRL [8] proposes an IRL method on nuPlan [9] which is the first real-world planning benchmark. nuPlan is still under active development and has not been fully released. This benchmark uses an offline perception system that does not need to run in real-time on the AV, can use future information, and is used to generate highly accurate automatic labels which are considered to be the ground truth for the nuPlan dataset [9].

**(2) Disentanglement of contributing factors:** To correctly understand the importance of different elements of a planning system, an independent analysis that excludes perception errors is valuable. For example, it would be hard to compare the importance of two input variables for PlanT if one variable has larger perception noise due to the specific choice of perception system. Our analysis can then enable better design for perception systems which focus on reducing errors for the most important variables.

**(3) Computational expenses of experiments:** The training and evaluation times are significantly shorter when assuming perfect perception. PlanT can be trained in 12 GPU hours on 2080ti GPUs, whereas the perception model used in the following experiment requires ~400 GPU hours on more expensive V100s. The runtime during evaluation also increases by roughly a factor of 3 with perception.

---

> ### Author Response · Authors · 2022-08-25
> **General response to all reviewers (2/2)**
>
>
> ## Additional results with perception module
> Here, we provide additional results for PlanT in combination with a perception module. For this, we use LiDAR and RGB sensors and leverage a TransFuser [10] backbone for sensor fusion in order to predict bounding boxes and the route. We observe that our proposed PlanT model together with this off-the-shelf perception module (“PlanT with perception module”) outperforms the current state of the art [10] on the Longest6 benchmark in terms of the main metric for CARLA, Driving Score (DS).
> We will add a detailed description of these additional results in the revised submission of our paper.
> &nbsp;
> | Model        | DS &#8593;  | RC &#8593; | IS &#8593;| CR &#8595;|
> | :------------- |-------------:| -----:|-----:|-----:|
> **Sensor-based** ||||
> | LAV [11] | 32.74±1.45 | 70.36±3.14 | 0.51±0.02 | **0.84±0.11** |
> TransFuser [10] | 47.30±5.72 | **93.38±1.20** | 0.50±0.60 | 2.44±0.64 |
> PlanT with perception module &nbsp;&nbsp;| **56.41±0.83** | 88.68±3.50 | **0.65±0.05** | 1.01±0.06 |
> **Privileged** ||||
> Rule-based |29.09±2.12 |38.00±1.64 |0.84±0.00 |0.64±0.07 |
> AIM-BEV [6] |45.06±1.68 |78.31±1.12 |0.55±0.01 |1.67±0.16 |
> Roach [7]  |55.27±1.43 |88.16±1.52 |0.62±0.02 |0.76±0.07 |
> PlanCNN |77.47±1.34 |**94.53±2.59** |0.81±0.03 |0.43±0.05 |
> PlanT |**81.36±6.54** |93.55±2.62 |**0.87±0.05** |**0.31±0.12** |
>
> &nbsp;&nbsp;
> ## Sparse route representation vs. full HD map
> In our paper we show that the route encodes enough information to reach expert-level performance in the CARLA environment, but we agree that the route is not the only relevant aspect of the environment for real-world autonomous driving. However, by showing the sufficiency of the route in the simulated setting, we argue that the route is particularly important to encode and should be focused on even in a more complex environment. Since our PlanT architecture can be theoretically extended by adding more tokens for different object types, encoding other parts of the map is possible and an interesting future direction when more complex benchmarks are available.
> We will update the corresponding claim in our revised submission (which will be uploaded shortly) to clarify this point.
>
> We hope we could clarify all questions and welcome further suggestions and comments to make our paper stronger.
>
> &nbsp;&nbsp;
> #### References:
> [1] Jaime Fernández Fisac , Eli Bronstein, Elis Stefansson, Dorsa Sadigh, S. Shankar Sastry and Anca D. Dragan. “Hierarchical Game-Theoretic Planning for Autonomous Vehicles.” In ICRA, 2019.
>
> [2] Bingyu Zhou, Wilko Schwarting, Daniela Rus and Javier Alonso-Mora. “Joint Multi-Policy Behavior Estimation and Receding-Horizon Trajectory Planning for Automated Urban Driving.” In ICRA, 2018.
>
> [3] Edward Schmerling, Karen Leung, Wolf Vollprecht and Marco Pavone. “Multimodal Probabilistic Model-Based Planning for Human-Robot Interaction.” In ICRA, 2018.
>
> [4 Dorsa] Sadigh, S. Shankar Sastry, Sanjit A. Seshia and Anca D. Dragan. “Planning for Autonomous Cars that Leverage Effects on Human Actions.” In RSS, 2016.
>
> [5] Rowan McAllister, Blake Wulfe, Jean Mercat, Logan Ellis, Sergey Levine, and Adrien Gaidon. ”Control-Aware Prediction Objectives for Autonomous Driving,” In ICRA, 2022.
>
> [6] Niklas Hanselmann, Katrin Renz, Kashyap Chitta, Apratim Bhattacharyya and Andreas Geiger. “KING: Generating Safety-Critical Driving Scenarios for Robust Imitation via Kinematics Gradients.” In ECCV, 2022.
>
> [7] Zhejun Zhang, Alexander Liniger, Dengxin Dai, Fisher Yu and Luc Van Gool. “End-to-End Urban Driving by Imitating a Reinforcement Learning Coach.” In ICCV, 2021.
>
> [8] Tung Phan-Minh, Forbes E Howington, Ting-Sheng Chu, Sang Uk Lee, Momchil S. Tomov, Nanxiang Li, Caglayan Dicle, Samuel Findler, Fráncisco Suarez-Ruiz, Robert E. Beaudoin, Bo Yang, Sammy Omari and Eric M. Wolff. “Driving in Real Life with Inverse Reinforcement Learning.” In ArXiv, 2022.
>
> [9] Holger Caesar, Juraj Kabzan, Kok Seang Tan, Whye Kit Fong, Eric M. Wolff, Alex Lang, Luke Fletcher, Oscar Beijbom and Sammy Omari. “nuPlan: A closed-loop ML-based planning benchmark for autonomous vehicles.” In ArXiv, 2021.
>
> [10] Kashyap Chitta, Aditya Prakash, Bernhard Jaeger, Zehao Yu, Katrin Renz and Andreas Geiger. “TransFuser: Imitation with Transformer-Based Sensor Fusion for Autonomous Driving.” In TPAMI, 2022.
>
> [11] Dian Chen and Philipp Krähenbühl. “Learning from All Vehicles.” In CVPR, 2022.

---

### Meta-Review · Area_Chair_zAYr · 2022-08-09

**Recommendation:** Accept (Poster)
**Confidence:** 4

**Metareview:**

The paper presents an end-to-end approach for planning that uses a transformer architecture to predict future waypoints. The paper lays emphasis on sparse object-centric representations and further attempts to extract explanations based on attention scores learned by the transformer architecture.

The reviewers find the paper well motivated and clearly written and the approach is simple
and easy to understand. The analysis of attention score correlates with human intuition in driving scenarios. However, the reviews express the concern that this work assumes perfect perception and hence the conclusion that object centric representations are enough to encode all relevant information (say in relation to approaches that take BEV as input) is strong and may not be well substantiated by the results in this paper. Further, the assertion related to the sufficiency of the driving route (in contrast to the whole map) needs further analysis to ascertain if the effect is observed due to the characteristics of the simulation setup.

During the rebuttal phase, the authors addressed a central issue of assuming perfect perception and provided results with an integrated off the shelf perception module. Further, the technical exposition of the proposed architecture was strengthened with inclusion of details on the waypoint prediction and auxiliary loss which improve the paper.  In relation to the assertion that the driving route (as opposed to the whole map) is sufficient for planning, the authors concur that the in general the route may not be encoding all the context needed for planning.

Based on the inputs from reviewers, I observe that the paper attempts to tackle several questions associated with learning planners for autonomous driving: whether object centric representations are better or worse than BEV-context techniques, the need for explainability at object level, how can transformer-inspired architectures can be used in this setting and how well do they work. Given the breadth of issues, it is challenging to clearly articulate the technical gap that is finally filled with this paper. Hence, I would strongly urge the authors to explicitly state the precise technical gap their work is trying to fill (somewhere between sections 2 and 3) which would help the reader to better situate the contribution of this work.

---

> ### Author Response · Authors · 2022-08-25
> **Response to Meta Review**
>
> We thank the Area Chair for the meta-review and all the reviewers for their valuable feedback and favourable impression. We address the concerns in our general comment, the individual responses as well as our revised submission of the paper (which will be uploaded shortly). We hope we could clarify all questions and welcome further suggestions and comments to make our paper stronger.
>
> We address the two main points of the meta-review in the general response to all reviewers including a discussion about (1) the assumption of perfect perception: we show that we can reach [state-of-the-art performance](https://openreview.net/forum?id=80vpxjt3vq&noteId=MgulWWjQJF-) on the Longest6 Benchmark and add a [detailed discussion](https://openreview.net/forum?id=80vpxjt3vq&noteId=BAK_iA8SMb9) justifying our assumption and (2) the [sparse route representation vs. full HD map](https://openreview.net/forum?id=80vpxjt3vq&noteId=MgulWWjQJF-).